# Possible Consequences of Climate Change on Survival, Productivity and Reproductive Performance, and Welfare of Himalayan Yak (*Bos grunniens*)

**DOI:** 10.3390/vetsci9080449

**Published:** 2022-08-22

**Authors:** S. Sapkota, K. P. Acharya, R. Laven, N. Acharya

**Affiliations:** 1School of Veterinary Science, Massey University, Palmerston North 4442, New Zealand; 2Animal Quarantine Office, Department of Livestock Services (DLS), Kathmandu 44600, Nepal; 3Department of Veterinary Medicine, University of Cambridge, Madingley Road, Cambridge CB3 0ES, UK; 4Department of Pharmacology and Toxicology, The University of Texas at Austin, Austin, TX 78712, USA

**Keywords:** yak, climate change, temperature, heat stress, diseases, welfare

## Abstract

**Simple Summary:**

Climate change is a global issue, with a wide range of ecosystems being affected by changing climatic conditions including the Himalaya. Yak are exquisitely adapted to the high-altitude conditions of the Himalaya and are thus highly likely to be affected by climate change. This paper reviews the evidence of how the reported impacts of climate change on the environment and ecosystem of the Himalaya are affecting the survival, productivity and welfare of Himalayan Yak. This review identified that we do not know how big the impact of climate change is on yak as very few papers have measured that impact and, in many cases, potentially climate-change-related effects (such as changes in feed supply) are principally driven by human factors.

**Abstract:**

Yak are adapted to the extreme cold, low oxygen, and high solar radiation of the Himalaya. Traditionally, they are kept at high altitude pastures during summer, moving lower in the winter. This system is highly susceptible to climate change, which has increased ambient temperatures, altered rainfall patterns and increased the occurrence of natural disasters. Changes in temperature and precipitation reduced the yield and productivity of alpine pastures, principally because the native plant species are being replaced by less useful shrubs and weeds. The impact of climate change on yak is likely to be mediated through heat stress, increased contact with other species, especially domestic cattle, and alterations in feed availability. Yak have a very low temperature humidity index (52 vs. 72 for cattle) and a narrow thermoneutral range (5–13 °C), so climate change has potentially exposed yak to heat stress in summer and winter. Heat stress is likely to affect both reproductive performance and milk production, but we lack the data to quantify such effects. Increased contact with other species, especially domestic cattle, is likely to increase disease risk. This is likely to be exacerbated by other climate-change-associated factors, such as increases in vector-borne disease, because of increases in vector ranges, and overcrowding associated with reduced pasture availability. However, lack of baseline yak disease data means it is difficult to quantify these changes in disease risk and the few papers claiming to have identified such increases do not provide robust evidence of increased diseases. The reduction in feed availability in traditional pastures may be thought to be the most obvious impact of climate change on yak; however, it is clear that such a reduction is not solely due to climate change, with socio-economic factors likely being more important. This review has highlighted the large potential negative impact of climate change on yak, and the lack of data quantifying that impact. More research on the impact of climate change in yak is needed. Attention also needs to be paid to developing mitigating strategies, which may include changes in the traditional system such as providing shelter and supplementary feed and, in marginal areas, increased use of yak–cattle hybrids.

## 1. Introduction

Domestic yak (*Bos (Poephagus) grunniens*) and their hybrids with domestic cattle (both *B. taurus taurus* and *B. t. indicus*) are hardy animals that are most commonly found in the highland regions of Asia including the Himalaya [1,2]. Yak are usually found in areas that are 3000 to 6000 m above sea level, and are adapted to extremely cold, harsh environments with low atmospheric oxygen concentrations and high levels of solar radiation [3]. They are used for a huge range of purposes including producing milk and milk products (cheese, butter, and whey), fuel (yak dung), meat, blood, fat, hair, and skins [4]. Yak are also used as a pack of animals, being able to travel up to 15 km per day in high-altitude areas [5] and carry loads of up to 100 kg in weight [5].

Like all *Bos* species, yak is a strict herbivore. They can graze/browse a wide range of vegetation from grass to small shrubs, grazing short forage like sheep (using their incisor teeth and lips), and browsing longer material like cattle (using their tongues to grasp plant material) [6]. Yak usually rely on natural vegetation as their feed source as supplementary feeding (other than wilted forages) is limited in most yak herding regions, including the Himalayas [7].

However, in traditional yak-farming areas, the growing season for natural vegetation is very short (e.g., 90–120 days on the Qinghai-Tibetan plateau) [8]. Temperature is the key factor limiting herbage growth, but rainfall during the warmer months also plays a crucial role [9]. This short grazing season has resulted in the main farming method used by traditional yak herders being transhumance pastoralism [10,11] with yak using high-altitude alpine pastures during summer and migrating to lower altitudes during winter. This migration also allows yak to encounter a relatively consistent ambient temperature throughout the year [3].

Despite this migration, from October to March, feed shortages are common due to thick snow layers over the winter pastures. During these shortages, wilted vegetation may be fed to yak, but, at best, this supplementary feeding is sufficient for maintenance only [12]. Thus during winter, yak usually lose body weight (up to 17–25%) and body condition and have an increased risk of mortality [13].

Yak are extremely well adapted to intense cold. They can survive ambient temperatures of −40 °C, and they perform best when the average annual temperature lies below 5 °C and the average temperature in the hottest month is less than 13 °C [6]. The key adaptations of yak to cold are: (1) a compact body shape with a short neck, limbs, and tail along with small ears resulting in a small surface area for heat loss; (2) a long thick fleece of outer hairs on the chest, legs, and flanks which traps air against the body; and (3) a thick but fine undercoat of downy hairs [12,14]. The outer hairs are produced by the primary hair follicles and the downy coat by secondary follicles; the balance between the two is highly seasonal, with significantly more secondary follicles being present in the cold seasons [14]. It is important to note that all of these adaptations are designed to reduce heat loss rather than increase heat production (which would be energetically difficult to achieve from winter forage). These adaptations, even with the seasonal loss of downy hair from secondary hair follicles, mean that yak are intolerant to heat stress. This is exacerbated by the lack of functional sweat glands with functional sweating being restricted to the muzzle region [15].

The reproductive performance of yak has a close relation to climate and season [16]. Although oestrus behaviour can be observed at most times of the year [7,17], yak are best considered as seasonal breeders (especially in areas such as the Himalaya) with conception and mating generally occurring during the warmer months of the year [16]. The start and end of the breeding season is determined by ambient temperature and relative humidity. These influence the availability of feed and fodder and thus the nutritional status of the yak. Nutrition is also the main factor influencing ovarian function (alongside a minor role for photoperiod) [18] The timing of the breeding season has a close relationship with factors that determine climate, i.e., the latitude, longitude, and altitude of the pastures where yak are kept, with breeding season varying markedly across the Himalaya [7,19].

The following three issues mean that yak are likely to be exquisitely sensitive to climate change (especially increasing temperatures in both warm and cold months): (1) adaptation to cold environments; (2) reliance on alpine herbage for nutrition; (3) the close link between climate and fertility. Thus, the aim of this review was to identify the possible impacts of climate change on survival, welfare, production, and reproductive performance of yak in the Himalayan region and some potential solutions to these issues.

## 2. Climate Change in the Himalaya

The Himalayan ecosystem has been significantly affected by climate change [20]. The rise in temperature has altered climate patterns, destabilized the patterns/forms of water resources [21], and increased the risk of natural hazards [22] as well as affecting the growing cycle of plants, and the migratory pattern of animals [23].

### 2.1. Temperature

Based on data from 478 meteorological stations across the Hindu-Kush Himalayan region collected between 1961 and 2015, there had been a significant decrease in the number of extreme cold events (cold nights, cold days, and frost days) and a concomitant increase in the number of extreme warm events (warm nights, warm days, and summer days) (see Table 1 below) [24]. Similar trends have been reported in the western Himalaya [25].

This increase in temperature is most marked in the winter season. Shrestha et al. [26] reported that, between 1982 and 2006, across multiple ecoregions in the Himalaya, there was a warming trend of 0.07 °C per year in the winter and 0.03 °C per year during the summer season. Current projection models suggest that this warming will continue; Shrestha et al. [27] estimated that, between 2020 and 2050, the average temperature across the Hindu-Kush Himalayan region will increase by 1–2 °C (and in some areas by 4–5 °C).

### 2.2. Precipitation

Average annual precipitation across the Himalaya increased by 163 mm (6.52 mm per year) between 1986 and 2006 [26]. This increase has been very seasonal, with an increase of 187 mm (7.48 mm per year) during the summer months but a decrease of 17 mm (−0.68 mm per year) during the winter months. Climate models suggest that precipitation will increase by an average of 5% (maximum 25%) between 2020 and 2050, with a longer and more erratic monsoon season with fewer but more intense extreme rainfall events [27]. This is likely to be particularly pronounced in the Eastern Himalaya, while the southern part will show a slightly decreasing trend in extreme rainfall events [27]. This shows that extremes of both rain and drought could potentially bring a challenge to the Himalayan ecosystem.

### 2.3. Vegetation

Temperature changes have meant that high-altitude, cold-adapted plant species have shifted to higher altitudes and been replaced by species that are better adapted to warmer temperatures [28]. This is evident in the shifting in the treeline by up to 0.37 m/year across the central Himalaya [29]. These changes have resulted in alterations in vegetation type, with alpine meadows and herbs being replaced by shrubs [30,31,32]. This trend is predicted to continue, with higher CO_2_ scenarios being associated with greater replacement of herbs by shrubs [32]. Shrub encroachment is probably the major climate-change-associated factor decreasing the availability and quality of grazing vegetation in the Himalaya [30,31].

The changes in precipitation patterns are also crucial. Variation in precipitation between years greatly influences the production of vegetation [33], with accumulated precipitation playing a key role in seasonal vegetation production in the Himalaya [33]. Although biomass reduction induced by insufficient precipitation will reverse after good rainfall/snowfall [34], the ability of the land to recover after water scarcity will gradually decrease over time [33]. The processes responsible for this decline in responsiveness include soil erosion, a decline in infiltration or moisture-holding capacity of the soil, loss of seed banks, and shrub encroachment [35]. These changes will result in a long-term continued decrease in forage biodiversity, availability, and quality, with Yang et al. [36] reporting that increasing shrub cover was associated with reduced total herbaceous forage production and with reduced crude protein intake by yak.

Extreme weather events such as floods, droughts, and high temperatures, the risks of which are increased by climate change, increase the likelihood of outbreaks of pests and other diseases in alpine vegetation, because they affect plant defence mechanisms and make them more susceptible to pests and pathogens [37].

The changes in temperature and rainfall patterns have resulted in an earlier start of the growing season and earlier bud outbreaks as well as increased germination rates [26]. The impacts are not solely beneficial. The changes in temperature and timings are especially beneficial for weed species which have become much more invasive, thus crowding out useful forage plants. Additionally, many of the forage species relied on by yak need snow cover to insulate them from the winter cold, so have been hit hard by the loss of that cover even though winter air temperatures have increased, while other species that require winter chilling for bud break may not get sufficiently low temperatures over a sufficiently long period for that to occur [38]. Even if we account for changes due to increased shrub cover, increased weed growth, and loss of forage during winter has meant that in many cases, the main impact of climate change on alpine yak pastures has been a reduction in yield and productivity, despite the apparent increase in the length of the growing season [39].

## 3. Yak Population

There is a lack of precise statistics in regard to the current population of domestic yak. According to Weiner [12], there are around 13–14 million domestic yak population in the world with majority of them being in China (Table 2). Outside of China domestic yak populations are declining in many countries, e.g., India, Bhutan, and Nepal [40,41,42]. In addition, increasing hybridisation has made it difficult to identify the population of pure domestic yak population.

## 4. Possible Consequences of Climate Changes on Yak

The thermo-neutral zone of yak ranges from 5–13 °C [6]. The risk of heat stress in yak can be assessed by measuring the temperature-humidity index (THI), which combines temperature and relative humidity. The yak’s physiological focus on reducing heat loss means that, compared to cattle, the THI threshold above which yak are likely to begin experiencing heat stress is much lower (52 vs. 72 for yak and cattle, respectively) [47]. If the relative humidity is 65%, then air temperatures >13 °C will result in a THI > 52. The change in climate in the Himalaya has meant that this THI threshold is being increasingly exceeded even during winter [48], thus putting yak at higher risk of becoming heat stressed.

The pastoral nature of yak farming, especially in its traditional transhumance form, means that artificial heat stress mitigating strategies, such as providing shelters, are difficult to provide in many circumstances. This means that heat stress due to climate change could potentially have severe impacts on the health and welfare of farmed yak. However, to date, there have been no published studies that have directly evaluated the effect of this heat stress on the productivity, disease risk, and immune functions of yak. However, the results of previous studies on yak and related domestic species, especially cattle, have suggested that the increase in the number of days where the THI of yak is exceeded is likely to have negative impacts on physiology, production, immune function, and disease risk.

### 4.1. Climate Change and Yak Physiology

The main effects of climate change on yak physiology are likely to be principally mediated through environmental temperature. Increased temperature leads to an increase in respiration rate as that is the yak’s main method of heat dissipation [6]. If this increased temperature continues, pulse rate and, eventually, the rectal temperature will rise [49,50].

Another significant physiological change associated with environmental temperature is plasma cortisol concentration. In yak, plasma cortisol concentrations are lower in warmer months than in cooler seasons [51]. The physiological significance of this cortisol reduction is not clear, but it suggests that one of the adaptive mechanisms of the yak to prolonged elevated heat loads is decreasing adrenal cortical output as found in cattle [52,53,54]. Alongside this decrease in cortisol concentrations, lower blood glucose, and volatile fatty acid concentrations are also observed during the warm-humid seasons compared to the cold-humid seasons [50]. These changes in blood metabolites are consistent with the findings in cattle that cold exposure causes blood glucose concentrations to rise in response to increases in circulating thyroid and adrenal hormones which contribute to metabolic heat production [55].

One crucial physiological response to heat stress is the response of the reproductive system. Yak reproduction is affected by temperature, as yak are much more likely to come into oestrus during the early morning or evening than in the hotter parts of the day, and they are more likely to be seen in oestrus on overcast rather than clear days [56]. However, the underlying physiology of this effect is unclear, as there have been no published studies on reproductive physiology in yak and heat stress.

Studies in cattle may provide some guidance as to the likely effect of heat stress on reproductive physiology in yak. In cattle, heat stress has been shown to have effects across the whole of the oestrus cycle, for example, decreasing luteinising hormone concentrations and depressing the LH surge [57,58], while also stimulating premature luteinisation and increasing progesterone production [58]. Heat stress also impairs oocyte development and disrupts normal follicular function [59,60]. If these effects also occur in yak, it is unlikely that they will be simple to ameliorate once they occur.

### 4.2. Climate Change and Reproductive Performance

Thus, the physiological changes associated with heat stress are likely to result in reduced reproductive performance. These impacts are likely to be exacerbated by the impacts of climate change on yak nutrition. As discussed earlier, nutrition is critically associated with yak reproduction. As the grass starts growing in May, the body condition of yak begins to improve, and in June, dry females begin to exhibit oestrus peaking around July–August. Lactating yak have a delayed return to good body condition and, thus, tend to show oestrus later (September to November) than dry yak, and are more likely to be non-pregnant at the end of the breeding season [56]. The changes in forage availability and quality as a result of climate change are likely to delay the recovery of body condition in both dry and lactating yak, increasing the proportion that is non-pregnant at the end of the breeding season (especially lactating yak). It is also important to note that a reduction in forage quality and availability is also likely to impact the onset of puberty which is closely linked to nutrition [61]. Thus, climate change is likely to have long-term effects on yak fertility unless there is greatly increased use of supplementation.

It has also been suggested that abortion in yak may be related to heat stress [62], but this suggestion was based on data from one farm where ~40% of abortions occurred in May/June. This farm had a much higher rate of abortion than is normal for yak, so it is unclear how representative it is, and the conclusion was based only on the timing of the abortions, and no investigation of the cause of the abortions seems to have been undertaken. More research is needed to properly evaluate the effect of heat stress on yak fertility.

### 4.3. Climate Change and Productivity

The changes in forage quantity and quality described earlier are likely to have a significant negative effect on yak productivity. Yang et al. [36] reported that the changes in forage quantity and quality associated with shrub encroachment led to reduced growth in yak. Similar results are not available for milk production and reproductive performance, but the impacts are likely to be similar.

The evidence of an impact of heat stress on the milk production of yak is limited, with no long-term studies of comparative production under different temperature and humidity conditions. Shikui et al. [63] reported that over the short term, yak produced more milk (~0.1 to 1 kg/day) on cloudy, cool days (6.7 to 9.3 °C) than on the preceding and following warmer, clear sunny days (12.5–13.5 °C). This was a small study (only 19 animals) over an approximately two-week period in June, on one farm, so much more data are required to properly establish the likely impact of heat stress on milk production by yak. Furthermore, the underlying cause of these changes is unclear, but it might be related to dry matter intake, which is negatively related to THI in cattle [64], although this reduction in food intake may not be the sole reason for the reduction in milk yield. In cattle, Gao et al. [65] reported that despite similar dry matter intake, heat-stressed cows produced less milk than pair-fed cows kept in thermoneutral conditions. They [65] suggested that THI-related impacts on mammary blood flow could be responsible for some of this reduction. The cattle data thus suggest that milk production will reduce in yak suffering from heat stress but as impact on cattle is dependent on the level of productivity, with higher-yielding cows being more susceptible to heat stress [66], yak-specific data are needed to properly assess the likely range of losses.

### 4.4. Climate Change and Infectious Disease Occurrence

Yak are susceptible to a large number of infectious diseases, most of which are also present in local cattle or sheep [6,67]. Losses from infectious diseases can be high in individual herds [67], but it is likely that the literature is biased toward investigations in problem herds as there are few studies on the prevalence and economic impact of infectious diseases across multiple herds and multiple regions.

On first principles, it is likely that climate change is increasing the exposure of yak to disease. Firstly, increased temperatures may increase contact between yak and cattle, particularly during the winter period. As a large proportion of the infectious disease recorded in yak is likely to come from cattle [6], this increased contact is likely to increase the exposure of yak to cattle diseases. Secondly, climate change may affect vector-transmitted disease prevalence by shifting the geographical range of vectors, increasing their reproductive efficiency, and by altering vector–host interactions [68,69]. These changes could increase the incidence of diseases that are already established and/or result in the introduction of new diseases. Finally, the increased periods of nutritional stress caused by the impact of climate change on forage quality and quantity are likely to increase the susceptibility of yak to disease. Nevertheless, the lack of baseline data on disease prevalence in yak means that it is likely to be difficult to identify whether climate change is actually altering disease prevalence. Koirala et al. [70] surveyed in the Mustang district of Western Nepal, interviewing 71 households on the effects of climate change on livestock, particularly their perception of the impact of climate change on livestock disease. Although yak is common in the Mustang district, most of the respondents did not have a yak, so the survey does not provide direct evidence of changes in the perception of disease risk in yak herders. However, many of the households did have Jhopa (yak/cattle cross) and because disease transmission from cattle to yak is likely to be a key source of disease in yak, increased disease in Jhopa is likely to be reflected in yak.

Koirala et al. [70] claimed that there was “strong evidence to suggest that climate change has affected the occurrence, distribution, and prevalence of livestock disease in Nepal”. However, of their 71 respondents, 38 reported that either livestock disease patterns had not changed or had decreased. This is despite 60/71 respondents reporting that they had observed climate changes. Additionally, Koirala et al. [70] stated that mixed feeding (i.e., feeding supplements or by-products in housed animals) had become much more common and that free grazing had reduced, changes which would have been expected to increase disease occurrence.

Increases in the prevalence of parasitic diseases have been commonly linked to climate change. For example, Li et al. [71] linked increases in besnoitosis on the Tibetan plateau to climate change. However, they did not report a consistent increase in seroprevalence between 2012 and 2017 (the highest prevalence was recorded in 2014) and as seropositive animals were very rare, the study lacked the power to detect a true trend. Thus, their suggestion of a link between climate change and besnoitosis was not based on the data from their study. Another parasitic disease, babesiosis, which is a prime cause of mortality of young and adult yak in native yak herding regions, seems to be spreading in the yak population, with He et al. [72] highlighting climate change as a potential cause of this spread (principally because it was increasing the area of yak habitat that was suitable to the tick species that are the vectors of the pathogenic *Babesia* spp.). However, they also identified nutrition, animal movement, human activities, and the change from local resistant *Bos indicus* cattle to exotic susceptible *Bos taurus* breeds as potential risk factors which could increase the spread of babesiosis. So as with besnoitosis, for babesiosis, we lack both good quality prevalence data and data linking increases in prevalence to climate change.

A more data-based study was that provided by Khanyari et al. [73], who modelled the dynamics of gastrointestinal nematodes in trans-Himalayan pastures. They concluded that more than 30 years of climate change in the region had not resulted in an increase in the transmission potential of gastro-intestinal nematodes on pasture but that, relative to rainfall, temperature had become a more important determinant of pasture infectivity. The results of this modelling process highlight the complexity of determining the impact of climate change on disease prevalence, especially in a data-poor environment.

One particular area where climate change may be affecting the prevalence of parasites in yak is through its impact on medicinal plants. Yak herders rely on medicinal plants found in the Himalayan pastures for controlling parasitic infestations (both ecto- and endoparasites) [74]. However, these naturally occurring medicinal plants have become less common in the Himalaya [32,75], due to previously discussed climate-change-induced effects such as increases in plant pests, changes in the flowering season, geographical range shifts and shrub invasion [28,30,31,37].

The effect of climate change on the prevalence of non-parasitic diseases, such as brucellosis and foot-and-mouth disease, is even less clear with very limited data. Most analyses of climate change and yak diseases suggest that climate change will lead to increased disease (e.g., [15,23]) without providing data, while most datasets of disease prevalence, e.g., Mortenson et al. [76] are single timepoint studies which provide no data, on their own, as to disease patterns in yak. As far as the authors are aware, the only published systematic study of temporal patterns in non-parasitic disease in yak is the meta-analysis undertaken by Zhao et al. [77] who analysed the prevalence of brucellosis in Chinese yak. They reported that the “incidence of brucellosis was higher”, with the pooled prevalence being higher after 2012 than before. However, while suggestive of an increase, this study is not conclusive. Zhao et al. [77] separated their pooled prevalence over time into three categories—before 2012, 2012 to 2016, and 2016 or later—with the three pooled prevalences being 5.8, 11.5, and 7%, respectively. Thus, the changes in prevalence over time were relatively small, and there was no clear pattern of a continuous increase. Additionally, their modelling did not rule out there being no effect of time on the prevalence of brucellosis (95%CI of their regression coefficient being −0.01 to 0.205; *p* = 0.075). Zhao et al. [77] suggested that the National Brucellosis Control Plan which was issued in 2016 was responsible for the decrease seen after that year, but their dataset is too small to make such a conclusion, and it is too small to allow their model to distinguish between the pre-2012 prevalence and the prevalence in the 2012–2016 time period. Thus, their [77] conclusion that “in China, yak brucellosis is reviving” seems to be based more on their perception of disease risk than on their data. They [77] linked the increase in brucellosis to increased stocking density (and thus increased exposure to brucellosis). They linked the increased stocking density to degradation of grazing areas, a change which has been clearly linked to climate change [78,79]. However, overgrazing has also been shown to occur because of human activity driving changes in land use, and it is likely that if degradation of grazing is causing an increase in brucellosis in yak, then these socio-economic changes are currently more important in driving change in disease risk than climate change [23,80].

The lack of data on disease in yak makes it very difficult to assess whether disease prevalence is increasing, let alone relating any such changes to climate change. This does not mean that significant changes in disease prevalence are not happening, rather that we lack the data to identify such changes. We strongly agree with the conclusion by Zhao et al. [77] that extensive epidemiological investigations need to be conducted on brucellosis in yak, except that we would extend that to include all of the key diseases in yak.

In this discussion about climate change, it is important to stress that the lack of access to proper veterinary facilities, advice and medications is currently having a major effect on the prevalence and impact of disease in yak in the Himalayan region. If climate change is changing disease risk (or is going to), this lack of access to veterinary care will significantly increase the impact of such changes, so improving access to veterinary support needs to be a key part of any plan to mitigate the impact of climate change on yak.

## 5. Overall Welfare Status

The impact of climate change on yak welfare is difficult to assess because there is a paucity of information on yak welfare. As far as the authors are aware, there are no peer-reviewed papers on the assessment of yak welfare, and there is only one non-peer-reviewed conference abstract [81] which reported the results of an assessment of yak welfare in three regions of Bhutan using an assessment program derived from the Welfare Quality system [82]. The paucity of studies on yak welfare assessment means we cannot directly assess the impact of climate change on yak welfare. Nevertheless, potential impacts of climate change, such as the rise in extreme weather conditions, increases in parasitic and other diseases, and increasing difficulty accessing water sources are all likely to have reduced welfare [6,15,22]. The climate-change-related issue that is likely to have the most impact on yak welfare is shortage of forage in yak rangelands, which has been recognised as a major concern in yak farming region [83]. However, forage shortages are a result of a complex range of factors, some of which may be related to climate change (e.g., reduced snow cover and increased grazing by cattle) and some of which are probably not (e.g., bans on burning rangelands and increased collection of *Cordyceps* (caterpillar fungus)). As the key causes of forage shortages vary across regions within the same country [83], it is likely that determining how much of the impact of such shortages on yak welfare is due to climate change rather than socio-economic factors will be difficult. In addition, although climate change is likely to be having a deleterious impact on yak welfare, other key issues such as lack of proper treatment facilities and effective medicines, limited knowledge and skill of stockpeople, continuation of traditional farming practices, and limited support from the governments to uplift and promote sustainable yak farming are likely to be as, if not more, important in determining yak welfare.

There is thus a clear need for more welfare assessments in yak. In particular, we need a robust, internationally recognised protocol for such assessment, and, using that protocol, we need to collect data on yak welfare on an ongoing basis in yak herding regions across the Himalaya, alongside the collection of data on the impact of climate change on yak’s environment and its resources.

## 6. Adaptation Strategies to Mitigate Adverse Effects of Climate Change on Yak

Increasing ambient temperatures, more erratic precipitation patterns, higher rates of natural disasters, growing disease risk, and disruption of food plant ecosystems are all likely to bring more challenges to yak in the Himalaya. Addressing such effects is likely to take significant global effort and require substantial research. So, in the short to medium term, developing strategies to mitigate the effects of climate change are likely to be an essential step in helping yak to adapt to their changing environmental circumstances. Table 3 summarises some potential mitigation strategies, but this is clearly a very large area of research that deserves more space than is available in the current review.

## 7. Conclusions

This review has highlighted the clear effects of climate change on temperature, precipitation, and vegetation in the Himalayan ecosystem. Climate change has destabilised the ecosystem by increasing the ambient temperature, altering seasonal and climatic patterns (increasing extreme hot and cold events), causing erratic rainfall events, increasing natural disasters such as landslides and flood, and causing the replacement of native forages with unpalatable shrubs. These climate-change-induced events have clearly affected the traditional transhumance farming system of Himalayan yak. It is clear that there is an increased risk of heat stress in yak because of the increase in the number of days in both summer and winter where maximum THI is above the optimal THI of yak, and that this heat stress is likely to negatively affect welfare, productivity, and performance. The alteration in feed supply resulting from climate change is also likely to be negatively affecting welfare, productivity, and performance, as is the increase in disease risk. However, we lack the data to quantify these effects, and for some of the changes (particularly in relation to feed supply), it is difficult to distinguish between changes occurring due to climate change and changes driven by socio-economic factors. Additionally, non-climate-related issues such as lack of infrastructure (including veterinary facilities,) and poor management are possibly more important causes of poor welfare, productivity, and performance than climate change, and if not improved will act to exacerbate any future impact of climate change. Some immediate steps such as building temporary sheds, using water sprinklers, increased planned use of supplements, improved breeding practices, more effective rangeland management including rejuvenation and controlled grazing, should be implemented on the ground level. However, these steps can only slow down the impact of climate change on yak. On the broad spectrum, climate change is a global issue and must be addressed globally.

## Figures and Tables

**Table 1 vetsci-09-00449-t001:** Trends of extreme temperature indices over the Hindu-Kush Himalaya region in the period 1961–2015. Source: Sun et al. [24].

Indicator Name	Definition	Trend (d/10 Years)
Cold nights	Days when T_min_ < 10th percentile	−0.977
Cold Days	Days when T_max_ < 10th percentile	−0.511
Warm nights	Days when T_min_ > 90th percentile	1.695
Warm days	Days when T_max_ > 90th percentile	1.239
Frost days	Annual count when T_min_ < 0 °C	−3.636
Summer days	Annual count when T_max_ > 25 °C	6.741

**Table 2 vetsci-09-00449-t002:** Recent estimates of domestic yak populations in Himalaya.

Country/Region	Yak Population (Year)	Reference
Tibet	4.9 million (not dated)	Song et al. [43] cited Zhang [44]
Bhutan	38,642 (2021)	[41]
India	57,570 (2019)	[40]
Nepal	65,406 (2020)	[42]
Pakistan	25,900 (2013)	[45]
Afghanistan	4600 (2015)	[46]

**Table 3 vetsci-09-00449-t003:** Summary of some potential strategies to mitigate the effect of climate change on Himalayan yak.

Type of Modification	Mitigation Strategies
Physical modification	➢ Planting trees to provide shelter and shade [83] could be utilised wisely by avoiding forest degradation. ➢ Mud pens and other temporary pens used during winter [84] could be modified and used in summer. ➢ Concept of water collection tanks used for agriculture and household in the Himalaya can be utilised for yak drinking [82,85] during water shortage.
Breeding selection	➢ Hybridization with hill cattle can produce an animal that is better adapted the rising ambient temperature and can utilise rangelands that are too high for cattle and too low for pure yak [86]. ➢ Dimjo chauri (hybrids from mating cattle bulls with female yak) adapt well to both high and low-altitude areas and are more adaptable than Urang chauri (hybrids from mating yak bulls with local female cattle) [87,88]. ➢ Selection of heat-tolerant yak will aid the identification of genes related to thermotolerance which, if incorporated into breeding programs, will produce more resilient genotypes [89].
Nutritional modification	➢ Increased use of supplements would replace the reduced quantity and quality of the forage available improving productivity and performance. Suggested feed sources include feed blocks, concentrates and silage as well as mineral blocks to tackle potential mineral deficiencies [90], with particular attention being paid to feeding during the winter season [91]. ➢ More active management of alpine pastureland is also likely to be beneficial. Renovation with high-quality forages (20–30% legumes and 70–80% grass is likely to be a cost-effective strategy [92]. This can either be achieved through re-seeding or through transplanting of roots and stems. This is likely to be most effective it is combined with regular weed control and well-regulated rotational grazing [88,92]. ➢ Planting of tree fodder alongside pasture grasses will increase the availability of feed when pasture growth is limited [93]. Increased research on local forage and pasture species could be extremely valuable as it would allow the development of high-yielding and nutritious local plant varieties.

## Data Availability

Not applicable.

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
