# Peer review of "Possible Consequences of Climate Change on Survival, Productivity and Reproductive Performance, and Welfare of Himalayan Yak (Bos grunniens)"

_vetsci, 2022, doi:10.3390/vetsci9080449_

Round 1

Reviewer 1 Report

Work touches on an important topic which is the impact of climate on animal welfare. It suggests universal solutions that can be used globally.

Simple Summary (line 15-21) - to be completed.

Line 32-57 - is an Abstract? Introduction?

I lack data on the population of these animals in specific regions of the Himalayas and observed climate change that may negatively affect welfare. Maybe the map/chart would be valuable for a given district/region.

Temperature data (2.1. Temperature) is described in 2017. Are the values the same in 2022? The data was collected from 1961 to 2015. For such a long period could the adaptation of the Yaks? If so, what parameters were about?

Chapter 3.2. Climate change and reproductive performance is written in general without providing specific data in recent years. I suggest thoroughly describe this important chapter.

References not adapted to the requirements of the Journal.

The effect of cortisol should be described more clearly (line 230-241).

References not adapted to the requirements of the Journal.

Is it possible to count how much a "repair program" for Yaks will cost. Such calculations would be an interesting summary.

Author Response

Please see the attachment. Attachment contains combined response to the reviewers

Reviewer 2 Report

The authors must write the simple summary, because in the current manuscript does not exist.

The authors must perform a thorough revision of the references. There are within the manuscript citations that are not included in the references. Also the authors must consult the guidelines in order to revise the section of references,

Author Response

Please see the attachment. The file contains responses for both reviewers
